# Effects of Pair Versus Individual Housing on Performance, Health, and Behavior of Dairy Calves

**DOI:** 10.3390/ani10010050

**Published:** 2019-12-25

**Authors:** Shuai Liu, Jiaying Ma, Jinghui Li, Gibson Maswayi Alugongo, Zhaohai Wu, Yajing Wang, Shengli Li, Zhijun Cao

**Affiliations:** 1State Key Laboratory of Animal Nutrition, College of Animal Science and Technology, China Agricultural University, Beijing 100193, China; liushuaicau@cau.edu.cn (S.L.); majiaying@cau.edu.cn (J.M.); maswayi@yahoo.com (G.M.A.); yajingwang_cau@163.com (Y.W.); lisheng0677@163.com (S.L.); 2Department of Animal Science, University of California, Davis, CA 95616, USA; lgreyhui@hotmail.com; 3State Key Laboratory of Animal Nutrition, Institute of Animal Science, Chinese Academy of Agricultural Sciences, Beijing 100193, China; wzh07128@163.com

**Keywords:** calf, pair housing, individual housing, behavior

## Abstract

**Simple Summary:**

In modern dairy farming systems, calves are often housed in individual pens or hutches, which results in less social interaction with their peers during the milk-feeding period. The aim of this study was to evaluate the effects of pair versus individual housing on performance, health, and behavior of dairy calves from the milk-feeding period to the first week after mixing. Results showed that pair versus individual housing had no effects on body weight, starter intake or average daily gain during the milk-feeding period, while pair housing increased the growth performance of calves during weaning and postweaning periods, and the beneficial effects of pair housing on growth faded after calves were mixed and moved to group housing. Paired calves showed higher diarrhea frequency only in week three. The behavior of calves was altered at different periods, including increased time spent in feeding, chewing and ruminating, and decreased self-grooming time, and a drop of non-nutritive manipulation for all calves after they were mixed and moved to group housing. We also found less social contact may lead to more non-nutritive manipulation.

**Abstract:**

The aim of this study was to evaluate the effects of pair versus individual housing on performance, health, and behavior of dairy calves. Thirty female Holstein dairy calves were assigned to individual (n = 10) or pair housing (n = 10 pairs). The results showed that both treatments had a similar starter intake and average daily gain (ADG) during the preweaning period. During weaning and postweaning periods, paired calves had a higher starter intake, and the ADG of paired calves continued to increase but calves housed individually experienced a growth check. Paired calves showed higher diarrhea frequency only in week three. The results on behavior showed that feeding, chewing and ruminating time increased, and self-grooming time decreased with age during weaning and postweaning periods, and paired calves spent less time feeding, standing and self-grooming but more time lying during this time. After mixing, feeding, and chewing and ruminating time continued to rise, and self-grooming time continued to decline for both treatments. All calves spent less time standing and non-nutritive manipulation after mixing, and previously individually housed calves tended to increase non-nutritive manipulation. These results showed that pair housing improved growth during weaning and postweaning periods and that calves altered their behavior at different phases. Less social contact may lead to more non-nutritive manipulation.

## 1. Introduction

Under natural conditions, calves are nursed by the dam and tend to have social interactions with their peers or other animals [1,2]. In modern dairy farming systems, however, calves are often housed in individual pens or hutches. Hence, they are less likely to interact with their peers or other animals during the milk-feeding period. 

Previous work has indicated that different housing systems (group versus individual housing) affect the performance and health of dairy calves. Some studies showed that compared with individual housing, group housing increased weight gains [3], starter intake [4] and hay intake of dairy calves [5]. Conversely, other studies showed no effects [6] or even negative effects on weight gain for group-housed calves [7]. Furthermore, respiratory diseases and diarrhea were reported to occur more frequently in group-housed veal calves [8]. On the contrary, Babu et al. [9] reported that rearing calves in a group resulted in a lower disease incidence. In other cases, health outcomes were similar between different housing systems [10]. The variability among studies may be related to differences in management (e.g., the number of animals per group, milk volume provided, duration of the feeding period, weaning method, and disease diagnosis). From a behavioral standpoint, weaning from a milk-based diet to a solid diet is one typical stressor faced by dairy calves, in which case, calves vocalize more (d 37 to 55) [11,12]. After weaning, calves are mixed with unfamiliar animals and moved to a novel environment, which may cause aggressive interactions (d 91 to 126) [13]. The stress resulted from weaning and mixing can negatively affect animal welfare [14]. Social housing during the milk-feeding period may have beneficial effects on behavior and cognition ability of calves even after they were weaned and mixed with unfamiliar animals in a group. Several studies have shown that social housing improved resilience to stress (d 51 to 53) [15], as well as increased competitive behavior (d 49 to 56) [16] and interactions (d 56 to 91) [17] after weaning. Furthermore, previous research has mainly clarified the effects of social housing on lying and feeding [17,18,19]. However, other behavioral responses, such as standing, chewing and ruminating, self-grooming, and non-nutritive manipulation have not been well characterized when calves were weaned and moved to group housing. The primary objective of this study was to compare growth, performance and health as well as evaluate the effects of paired versus individual housing on calves’ behavior when they were weaned (d 42 to 56) and moved to group housing (d 63 to 70). We hypothesized that paired-housed calves would have better performance than individually housed calves. 

When calves were weaned and moved to group housing, they experience changes in the way of feeding and management, especially when they are introduced to a different diet (from a milk-based diet to a solid diet or total mixed ration) and social environment. These changes may impact the behavior of calves. Overvest et al. [18] reported the day to day changes in lying and feeding during the weaning period (d 40 to 48), and Horvath and Miller-Cushon [14] described the day to day changes in standing time of calves mixed in a group (d 60 to 74 ± 5). However, how behavior would change from one period to another was still not clear. Therefore, the secondary objective focused on calf behavioral changes from the weaning period (d 42 to 56) to when they were mixed in a group (d 63 to 70). We hypothesized that calves would exhibit less socially affiliative behavior, such as self-grooming, which may be related to greater activity and exploratory behavior when calves were initially moved to group housing [14]. We also predicted that less non-nutritive manipulation would be observed after calves were moved to group housing, as non-nutritive manipulation often occurred among individually housed calves, especially during the milk-feeding period [20].

## 2. Materials and Methods

### 2.1. Animals and Treatments

This study was conducted at China Agricultural University’s Dairy Education and Research Centre (Datong, Shanxi, China) in 2016, in accordance with protocols approved by the Ethical Committee of the College of Animal Science and Technology, China Agricultural University, Beijing, China (No. 2016DR07). Thirty female Holstein dairy calves were collected from the end of March to the mid-April and were assigned to individual (n = 10) or pair housing (n = 10 pairs) based on birthdate and body weight (mean ± SEM; 43.5 ± 0.59 kg). The age difference between calves in the same pair was within 48 h. Only calves with successful passive transfer of immunity (mean ± SEM; 6.24 ± 0.09 g/dL), determined by clinical refractometer 24 h after birth, were included in the study (serum total protein ≥5.5 g/dL). 

### 2.2. Housing and Management

#### 2.2.1. Preweaning, Weaning, and Postweaning

All calves were born in a calving pen and separated from their dams within 1 h after birth and weighed. After that, calves were moved to a separate and clean straw-bedded nursery room adjacent to the calving facility. If the younger calf of a pair was born within 24 h of the older calf of the pair, then the two calves were moved directly to the experimental calf barn less than 200 m from the nursery room. If the younger calf of a pair was not born within 24 h of the birth of the older calf, then the older calf was kept in the nursery room until the younger calf of the pair was born, accepting an age difference within a pair of maximum 48 h. Calves were transferred to the calf pens by a cart; paired-housed calves were transferred together within 3 h to 5 h after the birth of the younger calf, whereas individually housed calves were transferred alone. 

Individually reared calves were kept in individual pens (1.5 m × 2.0 m), while paired-housed calves were provided twice the area (3.0 m × 2.0 m). Calf pens were located under a 3-sided (solid, 1.1 m in height), roofed shelter with a metal gate at the front. Calves could hear calves and see calves in neighboring pens through the openings in the gate. Openings provided access to buckets (10 L for each one) placed 35 cm apart in the center of the pen for water and starter. Calves housed individually had two buckets (one for water, one for starter), while pair housing calves were provided twice the feeding facilities. All calves had free access to water and pelleted starter feed. All the feeding facilities were cleaned daily. The interior of each pen was bedded with sand and bedding was replaced weekly.

Colostrum was heated to 39 °C in a water bath. After that, colostrum was transferred to 4-L esophageal tubing bottles and fed to the calf through a tube within 2 h after the calf was born on d 1. From d 2 to 56, pasteurized waste milk (nonsaleable milk) was provided 3 times daily at 08:00, 15:00 and 20:00 and the volume of milk for each time was equal. During the preweaning period, calves were fed 6 L/d from d 2 to 7, 7.5 L/d from d 8 to 42. Weaning was carried out by reducing milk volume on d 43 and calves were fed 6 L/d from d 43 to 49 and then 3 L/d until d 56. At each milk feeding, the buckets for water were removed temporarily and milk buckets (5 L for each one) were placed in the same position of the pen. For each pair, two milk buckets were used at each milk feeding, whereas calves housed in individual pens had a single milk bucket. Milk buckets were cleaned after each feeding. After weaning, calves remained in their pens during the postweaning period (d 57 to 63 ± 1). No forage was offered before mixing. 

#### 2.2.2. Mixing Period

On d 64 ± 1, individual calves were mixed with the paired-housed calves according to the age and moved to the calf barn. There were 5 groups and each group consisted of 6 calves: 2 previously housed in individual pens and 2 pairs previously housed in pairs. The age difference between calves in the same group was within 48 h. The back wall of the group pen (5.0 m × 4.0 m) was solid with two sides made from horizontal tubular metal bars (bar diameter: 5.0 cm; distance between bars: 12 cm) and a neck rail at the front. The length of the neck rail allowed all 6 calves to eat simultaneously (83 cm per calf). Total mixed ration (TMR) was delivered twice daily at 10:00 and16:00. Each group pen was equipped with one automatic water trough (length: 120 cm, width: 40 cm, height: 70 cm, depth: 20 cm) and calves had free access to water. Sand was used as bedding material and was replenished when the group was moved on d 70. Three Digital Thermometers (Deli Electronic Commerce Co., Ltd., Ningbo, China) were spaced evenly and mounted above (1.0 m) the sand bedding in the calf barn to record daily temperature and humidity (maximum and minimum).

The temperature and humidity fluctuated according to weather conditions (mean ± SD; 14.7 ± 9.3 °C and 23.2 ± 10.1% relative humidity).

### 2.3. Sample Collection

#### 2.3.1. Feed Sampling 

Feed samples were collected weekly and immediately frozen at −20 °C until they were further analyzed. The nutritional composition (Table 1) of the dry matter, crude protein, neutral detergent fiber, acid detergent fiber, ether extract, crude ash, calcium and phosphorus were analyzed following the methods of AOAC International [21]. Throughout the study, starter intake was recorded daily based on the amount offered and refused by each calf from d 5 to 63 ± 1. TMR intake during the mixing period was not measured because of group housing. 

#### 2.3.2. Body Measurements and Blood Sampling

Body weights (BW) were measured weekly (d 1, 7, 14, 21, 28, 35, 42, 49, 56, 63, 70). Body length (shoulders to pins), withers height, hip height, and heart girth were also recorded at the same time points. Blood samples were collected via jugular venipuncture using vacutainer serum collection tubes containing no anticoagulant 24 h after birth. The blood samples were then centrifuged at 3500× *g*, 4 °C for 15 min. Serum total protein (TP) was determined by an optical refractometer (Honneur Nutritional Technology Co. Ltd., Beijing, China).

### 2.4. Health Check and Treatment

The health check consisted of three parts: (1) fecal scoring, (2) clinical examination of the respiratory system, (3) rectal temperature. Fecal scores were recorded daily at 10:00 each day until d 63 based on a 1 to 4 system according to the guidelines outlined by Larson et al. [22]. Scores were, 1 = firm, well-formed (not hard); 2 = soft, pudding-like; 3 = runny, pancake batter; and 4 = liquid, splatters, pulpy orange juice. Fecal score data were collected by one independent trained observer. All fecal scores were recorded by observing fecal matter on the ground of the pen or the tail and hindquarters of the calf. Fecal scoring was not conducted during the mixing period, because the fecal scores of an individual could not be accurately identified due to group housing. A diarrheic day was defined when the fecal score was >2. Weekly diarrhea frequency was calculated with the following equation: Diarrhea frequency = [(number of diarrhea calves × days of diarrhea) / (total number of calves × days of trial)] × 100%. Any calf with a fecal score >2 was treated according to the protocols established by the farm veterinarian (e.g., by administering antibiotic drugs and electrolytic solutions). Respiratory health was checked before each morning feeding through visually inspecting nasal discharge and listening to breathing difficulties with auscultation by the farm veterinarian and a member of the research team before morning milk feeding. If calves had signs of respiratory disease such as nasal discharge, cough and breathing difficulties, and a rectal temperature ≥39.5 °C, they were treated using an *Andrographis paniculata* injection (10 mL; Dazheng Tec-Phar. Co. Ltd., Changchun, China) for a maximum of 48 d; if respiratory disease or pyrexia was not alleviated, the calf received antibiotics treatment for a maximum of 48 d. Electrolytes were also administered intravenously to calves that had a severe respiratory disease until fully recovered. Throughout the study, one calf from individual housing was treated for 3 d during the mixing period because of nasal discharge and breathing difficulties, and no other calves had respiratory disease.

### 2.5. Behavioral Observations 

A digital color camera (DS-7800, HIKVISION, Hangzhou, China) was placed above each selected pen (placed 2.5 m in front of the pens and 3.5 m from the pen floor), monitoring the behavior of the calves. During nighttime hours (from 17:30 to 07:30), the infrared monitoring function of the camera would turn on automatically. The recorded behaviors (Table 2) included feeding, chewing and ruminating, lying, standing, self-grooming, non-nutritive manipulation, and social contact. 

Six individually housed calves and 6 pairs of paired-housed calves were selected randomly for behavioral observations from d 43 to 70. Based on previous results [18,23,24], the sample sizes of behavior variables were estimated to obtain a power of 0.8 under a significance level of 0.05. During weaning and post-weaning periods, the behavioral data were recorded for 48 h on d 43, 50, and 57. For the mixing period, the behavior data were recorded for 48 h on the second day of mixing (d 65 ± 1) to avoid the effects of transition stress on calves. In order to clearly identify each selected calf from the groups during the mixing period, all selected calves were photographed from the front, back, left, right, and above. The observer could record behavior based on each calf’s unique photos. For every 24 h duration (144 h in total for each calf), instantaneous scan-sampling with 5 min intervals was used to collect the lying, feeding, standing and chewing and ruminating data and continuous recording was used to collect the self-grooming, non-nutritive manipulation, and social contact data [25]. All behavioral data were recorded by one observer.

### 2.6. Statistical Analysis 

#### 2.6.1. Starter Intake, Growth, and Health Data

Throughout the study, data were analyzed at the pen level (based on a single calf per pen for the individual treatment and the mean of the 2 calves per pen in the pair treatment). Starter intake data were averaged by the week, except for the first week data, which were averaged across the last three days (d 5 to 7). Continuous variables with repeated measurements, including starter intake, average daily gain (ADG), BW, and structural growth, were tested for normality using the UNIVARIATE procedure of SAS (version 9.2, SAS Institute Inc., Cary, NC, USA). These data were then analyzed from week 1 to week 10 (as a whole) using the MIXED procedure of SAS. The model included the fixed effects of time, treatment, and time × treatment interaction and the random effect of pen To account for the repeated measures within-subject, the covariance structures were chosen for each repeated variable on the basis of best fit which was determined from the Bayesian information criterion. The heterogeneous first-order autoregressive structure was selected for starter intake, BW, and structural growth data, and for ADG data, the first-order autoregressive structure was selected. Data for fecal scores were summarized by the week and analyzed using the Chi-squared test.

#### 2.6.2. Behavioral Data

Behavioral data obtained for individual calves from video were also averaged by pen (a single calf per pen for the individual treatment and the mean of the 2 calves per pen in the pair treatment) across the 48 h in each observation week (week 7 and week 8 during weaning, and week 9 during postweaning and 10 during mixing). For each 48 h behavioral observation period, the average duration of each kind of behavior per 24 h was calculated. Behavioral data were analyzed separately by two stages: (1) weaning and postweaning, and (2) mixing. The comparison of social contact between two treatments was only analyzed during the mixing period, as individually housed calves had no social interaction before mixing. For stage 1, the effect of housing on behavior was tested using the MIXED procedure of SAS. The model included the fixed effects of treatment, week, and week × treatment interaction, and the random effect of pen. To account for the repeated measures within-subject, the first-order autoregressive structure was chosen for each behavior on the basis of best fit, which was determined from the Bayesian information criterion. For stage 2, the effect of housing on behavior was tested using one-way ANOVA. Lying, standing, non-nutritive manipulation and social contact data were normally distributed. Behavioral data of feeding, and chewing and ruminating were analyzed after logarithm transformation, and self-grooming data were analyzed after square root transformation to meet the normality assumption. The transformed data were back-transformed to report. 

All data were reported as least squares mean. Differences of *p* < 0.05 were considered significant and 0.05 ≤ *p* < 0.10 was considered a tendency.

## 3. Results

### 3.1. Starter Intake and Growth 

As shown in Figure 1, starter intake showed an upward trend over time (*p* < 0.001) for both individually and pair housed calves with no difference in starter intake between treatments during the preweaning period (*p* > 0.05). During weaning and postweaning periods, starter intake tended to be higher for paired-housed calves during week seven (860.0 vs. 658.1 ± 80.1 g/d, *p* = 0.09), and than for individually housed calves during week eight (1461.4 vs. 1123.1 ± 97.0 g/d, *p* = 0.02) and week nine (2237.4 vs. 1899.5 ± 113.5 g/d, *p* = 0.04). 

ADG increased over time (*p* < 0.001) for both treatments and no differences were found between treatments during the preweaning period (*p* > 0.05, Figure 2). During weaning and postweaning periods, the weight gain of paired-housed calves continued to increase, but individually housed calves experienced a growth check. The ADG for paired-housed calves tended to be higher during week seven (0.94 vs. 0.71 ± 0.07 kg/d, *p* = 0.08). Individually housed calves had higher ADG than calves housed in pairs during the mixing period (1.20 vs. 0.85 ± 0.09 kg/d, *p* = 0.01).

Throughout the study, the housing system (paired vs. individual) had no effects on BW (*p* = 0.50) and structural measurements (Table 3), including withers height (*p* = 0.55), heart girth (*p* = 0.38), abdominal girth (*p* = 0.14), and body length (*p* = 0.23). 

### 3.2. Health

Throughout the study, one calf from individual housing during the mixing period suffered from respiratory disease, and no other calves had respiratory disease. Diarrhea frequency is shown in Figure 3. Pair housing increased diarrhea frequency in comparison with individual housing of calves during week three (18.0% vs. 6.0%, *p* = 0.03), yet no differences were found between treatments in other weeks (*p* > 0.05). 

### 3.3. Behavior

As shown in Figure 4, during weaning (week 7–8) and postweaning (week 9) periods, feeding time increased (*p* < 0.001) for both treatments. Overall, individually housed calves spent more time feeding (83.0 vs. 53.1 ± 1.15 min/d, *p* = 0.04) compared with paired-housed calves during this period. After mixing, feeding time decreased for individually-housed calves but increased for paired-housed calves, and the previous housing system had no effects on feeding time after mixing (*p* = 0.82). Ruminating time increased over weaning and postweaning periods (*p* < 0.001) for both treatments and individually-housed calves tended to have greater ruminating time than paired-housed calves during week seven (2.56 vs. 1.79 ± 0.26 h/d, *p* = 0.09). After mixing, ruminating time continued to increase with age for all calves with no differences found between treatments (*p* = 0.61). 

Lying time increased during the weaning period and decreased during the postweaning period for all calves. Standing time increased during the postweaning period for all calves. Calves housed in pairs spent more time lying (17.3 vs. 16.4 ± 0.27 h/d, *p* = 0.03) and less time standing (6.33 vs. 7.11 ± 0.18 h/d, *p* = 0.01) compared with calves housed individually during weaning and postweaning periods. After mixing, lying time remained stable and standing time decreased for all calves, and the previous housing system had no effect on lying (*p* = 0.56) and standing time (*p* = 0.84). There was a decrease in self-grooming time for both treatments over weaning and postweaning periods (*p* = 0.01), and calves housed individually exhibited more self-grooming than calves housed in pairs (40.7 vs. 20.6 ± 4.10 min/d, *p* = 0.02). After mixing, self-grooming time continued to decrease for all calves, with no differences between treatments (*p* = 0.65). In addition, non-nutritive manipulation time did not change with calf age during weaning and postweaning periods (*p* = 0.62), and treatment had no effects on non-nutritive manipulation time during this period (*p* = 0.10). After mixing, all calves decreased non-nutritive manipulation time, and the non-nutritive manipulation time tended to be longer for calves that were previously individually-housed (33.0 vs. 16.9 ± 5.62 min/d, *p* = 0.07). During the mixing period, the previous housing system had no effect on social contact (Figure 5; *p* = 0.53).

## 4. Discussion

### 4.1. Starter Intake and Growth

During the preweaning period, we did not observe any differences in starter intake, ADG or BW between treatments, yet during weaning and postweaning periods, pair housing improved growth performance. Our results were consistent with previous research that reported increased starter intake [6,26] and ADG [23] during the weaning period for paired-housed calves. Such improvements are likely due to social facilitation [12] and social learning [27], which allow calves housed in pairs to learn faster and eat more. Local enhancement is another factor affecting the feeding of calves, through which the behavior of one calf draws the attention of another in the same pair toward a particular food source [28,29]. In addition, paired calves might experience a lower level of stress during the weaning period because of social buffering [12]. The social buffering benefits of early pair housing have been discussed recently by Overvest et al. [18], who demonstrated that social housing might improve the ability to cope with the weaning stress via the positive effects on feed acceptance and behavioral flexibility. During the postweaning period, we observed greater starter intake in pair housing. Similar results were reported by Pempek et al. [20], who also attributed it to social facilitation. Besides, the competitive feeding environment among paired calves may also have resulted in more starter intake during the weaning and postweaning periods, as calves may increase the rate of feed intake in the competitive feeding environment [17]. Our results contribute to a body of evidence indicating that pair housing is particularly beneficial to solid-feed intake, growth, and supporting a smooth transition at weaning [3].

After mixing, calves were grouped together. Warnick et al. [30] and De Paula Vieira et al. [12] reported that calves previously housed in groups or pairs gained more than those previously housed in individual pens when they were mixed and placed together. Some studies attributed these results to the beneficial effects of social housing, such as reduced neophobia to new ration [31,32] and greater competitive success [16] when mixed with unfamiliar animals. On the contrary, we found that previously paired-housed calves had less ADG after mixing compared with calves housed individually, and the final BW was similar between treatments. Somewhat interestingly, Miller-Cushon and DeVries [4] reported that though paired calves had greater performance during the weaning period (d 39 to 49), previous housing (paired vs. individual) had no effect on DMI, ADG or final BW once previously individually housed calves were paired with unfamiliar calves after weaning (d 50 to 84). Similar results were reported by Overvest et al. [18], who also demonstrated that once calves previously housed individually were paired after weaning (d 49 to 56), they exhibited more feeding time and thus increased their solid feed DMI to a greater extent over time than paired-housed calves, and eventually resulted in similar DMI between treatments. These results suggested that previously individually housed calves could get the same performance (e.g., DMI, BW, and ADG) through modifying feeding behavior after they were exposed to social housing with unfamiliar calves. In this study, all calves experienced a sudden feed transition to TMR, in which case, the beneficial effects of social housing on food neophobia may be weakened by transition stress. Furthermore, we observed similar feeding time between treatments during the mixing period. Thus, we speculated that the higher ADG in calves previously housed individually may result from higher feeding rate, allowing them to consume sufficient TMR to meet or exceed their nutritional requirements and finally compensating for a previously lower starter intake. Further work to address this possibility is encouraged.

### 4.2. Health

In the current experiment, diarrhea frequency for calves housed in pairs was higher than that for calves housed individually in week three, yet no differences were found in other weeks. Some studies reported that housing calves in groups exhibited more health problems owing to higher levels of infectious agents and calf-calf transmission [33,34]. On the contrary, others reported a lower incidence of diarrhea for calves housed socially [9], and some found no differences in incidences of diarrhea and respiratory problems [35] between paired-housed calves and individually housed calves. The various results indicated that health problems were not consistently associated with social housing. The incidence of disease relies on many factors including calf immunity, environment management, disease diagnosis, and the ability of a calf to cope with stress [36]. These factors rather than the housing system may play a critical role in inducing health problems. Greater health problems in a group housing system may also stem from the difficulty of detecting disease in groups [23]. There is not enough evidence to support a diarrhea-increasing effect of pair housing, thus the higher incidence of diarrhea in pair housing in week three was probably due to low immunity to infection of calves aged from two to four weeks [37] and individual differences.

### 4.3. Behavior

Limited research has described how behavior would change at different periods from weaning to mixing, or the effects of paired or individual housing on behavior during these periods. Our results suggested that all calves experienced behavioral changes from weaning to mixing including increased feeding and ruminating time, and decreased self-grooming time. 

The increase in feeding time and chewing and ruminating time over the weaning and postweaning periods aligned with the increase in feed intake. Besides, paired calves spent less time feeding but still had higher starter intake during weaning and postweaning periods, likely due to the competitive feeding environment as we discussed on starter intake and growth. Miller-Cushon et al. [17] found that calves housed in a competitive feeding environment had less time of feeding but an increased rate of feed intake compared with those housed in a noncompetitive feeding environment. Hence, paired-housed calves might increase their feeding rate rather than feeding time to consume more starter. 

In the present study, lying time declined while standing time rose during the postweaning period, which could be attributed to the increase in feeding time with increasing age during this period. The previous study [18] suggested that calves may change their lying behavior to accustom themselves to feeding behavior. In addition, calves housed individually exhibited more lying time than paired calves during weaning and postweaning periods, which is contrary to previous studies [23,38]. Previous research [20] also mentioned no effects of individual vs. paired housing on lying. The variant space allowance for calves among studies may be responsible for the discrepancy in results, as space allowance was a vital factor for the expression of normal behavior [23,39]. Further research is encouraged to study the relationship between space allowance and lying. After mixing, standing time decreased for all calves. Previous studies reported that calves were more active and moved more followed by a reduction in activity after the first day of introduction to a group [14,29], and calves had diminished behavioral reactions after the first 24-h period following regrouping [40]. Thus, we speculated that calves might not be as active as the first day of introduction to a group as the behavior data were recorded for 48 h on the second day of mixing to avoid the effects of transition stress on calves in this study. 

Self-grooming is expressed by calves as caring for their own body, and this behavior may be a means of satisfying socialization [27]. More self-grooming activities were observed in individually-housed calves in the present study, which was consistent with previous research [27], as the socialization was absent in these calves. In addition, self-grooming can also be an expression of stress. Taking rodent as a research model, previous studies [41,42,43] has reported that the relationship between stress and self-grooming can be described as an inverted U-shaped: Self-grooming typically occurs spontaneously at low stress and becomes longer during moderate stress and can be inhibited by high-stress states. Thus, the higher self-grooming of individually housed calves may respond to the higher stress (moderate stress) they faced compared with paired calves during weaning and postweaning periods. 

Non-nutritive manipulation commonly occurs within artificial rearing systems [44], which can be strengthened by social deprivation [45]. In the current study, a drop in non-nutritive manipulation time for all calves after mixing was observed, which was likely due to more social interactions among calves after mixing. Bokkers and Koene [46] also indicated that less social interaction was an important factor causing dairy calves to lick objects (a nonnutritive manipulation behavior). In this study, no effects were found on non-nutritive manipulation during the weaning or postweaning period, whereas individually-housed calves tended to spend more time on non-nutritive manipulation compared with paired calves during the mixing period, which was similar to the previous study [47]. Although the effect of the previous housing system on social contact was not significant during the mixing period, calves housed in pairs previously still exhibited more social contact numerically, which may result in less non-nutritive manipulation. 

## 5. Conclusions

Paired versus individual housing had no effects on body weight, starter intake or ADG during the preweaning period, while pair housing increased the growth performance of calves during weaning and postweaning periods, and the beneficial effects of pair housing on growth was weakened after mixing. Paired calves showed higher diarrhea frequency only in week three. Calves altered their behaviors at different periods from weaning to mixing, including increased feeding time and chewing and ruminating time, and decreased self-grooming time, and a drop of non-nutritive manipulation for all calves after mixing. Furthermore, less social contact may result in more non-nutritive manipulation.

## Figures and Tables

**Figure 1 animals-10-00050-f001:**
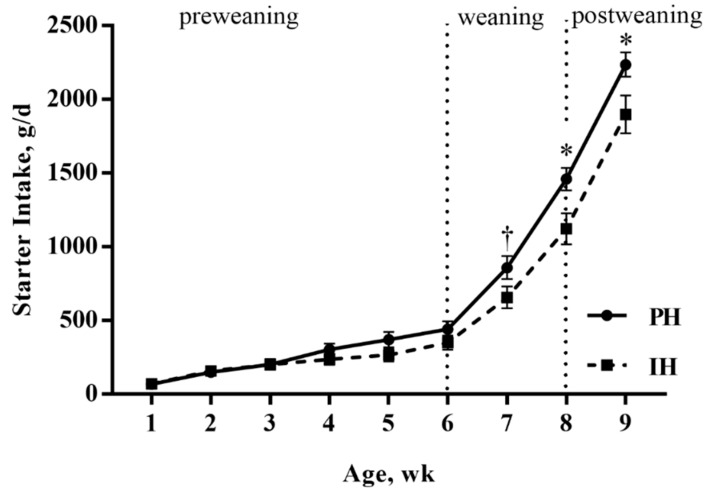
Starter intake (LSM ± SEM) for calves housed individually (n = 10 calves) or in pairs (n = 10 pairs) before mixing. PH = calves housed in pairs; IH = calves housed individually; wk = week. TMR intake was not measured because of group housing during the mixing period (week 10). * *p* < 0.05, † *p* < 0.10.

**Figure 2 animals-10-00050-f002:**
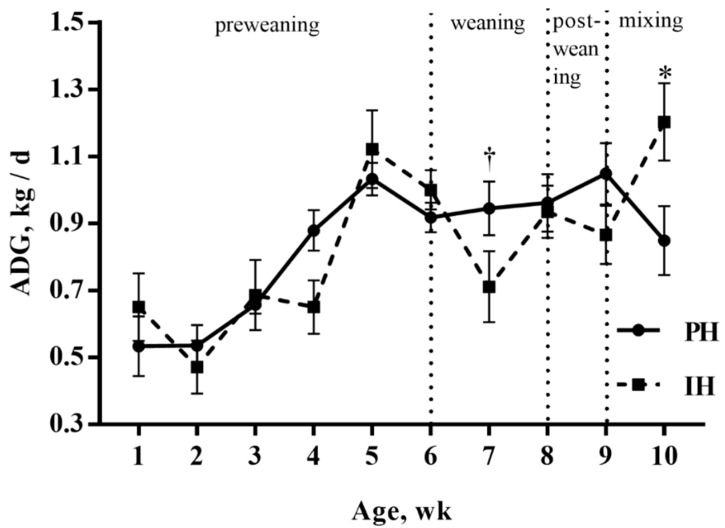
Average daily gain (LSM ± SEM) for calves housed individually (n = 10 calves) or in pairs (n = 10 pairs). PH = calves housed in pairs; IH = calves housed individually; wk = week. *p*-value: 0.90 (treatment), < 0.001 (week), 0.08 (treatment × week). * *p* < 0.05, † *p* < 0.10.

**Figure 3 animals-10-00050-f003:**
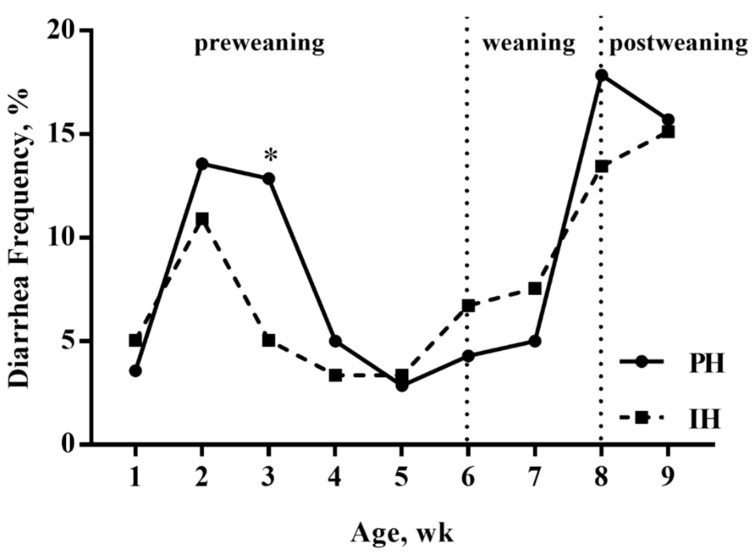
Effects of the housing system on diarrhea frequency before mixing for calves housed individually (n = 10 calves) or in pairs (n = 10 pairs). PH = calves housed in pairs; IH = calves housed individually; wk = week. Fecal scoring was not conducted during the mixing period (week 10), because the fecal scores of an individual could not be accurately identified due to group housing. * *p* < 0.05.

**Figure 4 animals-10-00050-f004:**
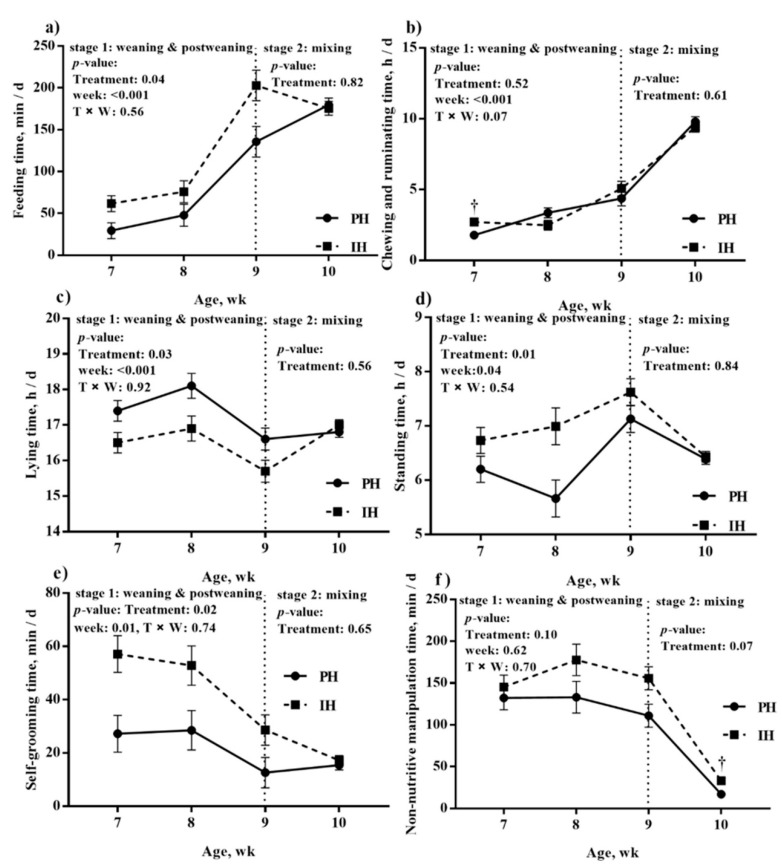
Effects of housing system on (**a**) feeding; (**b**) chewing and ruminating; (**c**) lying; (**d**) standing; (**e**) self-grooming; and (**f**) non-nutritive manipulation for calves housed individually (n = 6) or in pairs (n = 6). Stage 1 = from weaning to postweaning period, including weeks 7, 8, and 9. Stage 2 = mixing period, including week 10. PH = calves housed in pairs; IH = calves housed individually; wk = week. * *p* < 0.05, † *p* < 0.10.

**Figure 5 animals-10-00050-f005:**
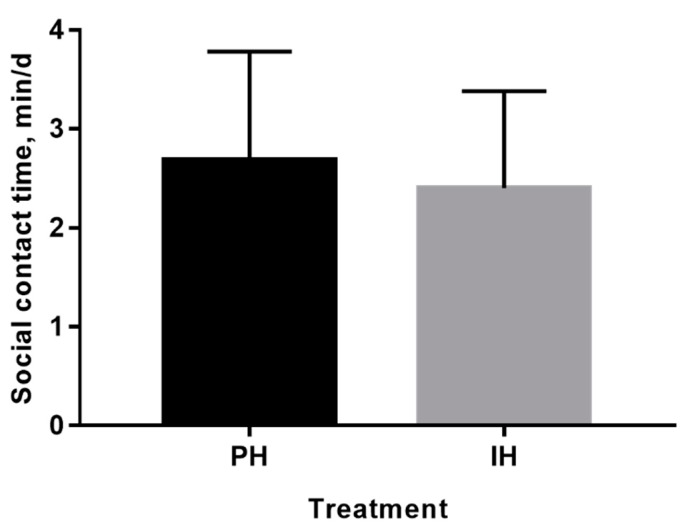
Effects of the housing system on social contact for calves housed individually (n = 6) or in pairs (n = 6) during the mixing period. PH = calves housed in pairs; IH = calves housed individually.

**Table 1 animals-10-00050-t001:** Nutrient compositions of milk, starter, and total mixed ration.

Nutrient Composition (%) ^1^	Milk	Starter	Total Mixed Ration ^2^
Dry matter (DM)	13.9	89.0	56.0
Crude protein, DM basis	3.30	22.5	13.6
Ether extract, DM basis	3.70	3.04	3.01
Crude ash, DM basis	9.0	6.07	12.8
Calcium, DM basis	0.60	0.91	-
Phosphorus, DM basis	0.60	0.52	-
Neutral detergent fiber, DM basis	-	15.0	36.0
Acid detergent fiber, DM basis	-	6.08	20.4

^1^ The nutritive values are the means of the results of the analysis of samples collected each week. ^2^ Contained steam-flaked corn (33.5%), alfalfa hay (21.2%), oat hay (21.2%), soybean meal (19.7%), and premix compound (0.4%) on a DM basis.

**Table 2 animals-10-00050-t002:** Ethogram of the recorded behaviors.

Behavior ^1^	Description
Standing	Standing with all four feet on the ground either active or inactive
Lying	Lying on the sternum with head held in a raised position or down
Feeding	Head in bucket accompanied by chewing movements, including milk drinking
Chewing and ruminating	Irregular, repetitive chewing without discernible food in the mouth
Self-grooming	Movements with tongue over own body surface
Non-nutritive manipulation	Biting, sniffing, sucking or licking pen structures; may include bucket if milk is not available
Social contact	One calf’s head was in contact with any part of the other calf including licking and sniffing of the other calf

^1^ If one calf exhibit multiple behavior at one time point, then the multiple behavior were all recorded. Social contact was recorded for calves during the mixing period.

**Table 3 animals-10-00050-t003:** Least squares mean of structural measurements and BW for calves housed individually (n = 10 calves) or in pairs (n = 10 pairs) from week 1 to week 10.

Item	Treatment ^1^		SEM	*p*-Value		
	PH	IH		Treatment	Time	Treatment × Time
Body weight, kg	69.8	68.7	1.12	0.50	<0.001	0.32
Withers height, cm	86.6	86.3	0.35	0.55	<0.001	1.0
Heart girth, cm	95.7	95.3	0.47	0.38	<0.001	0.90
Abdominal girth, cm	101.4	100.2	0.73	0.14	<0.001	0.40
Body length ^2^, cm	78.7	78.2	0.36	0.23	<0.001	0.85

^1^ PH = calves housed in pairs; IH = calves housed individually. ^2^ Body length was measured from shoulders to pins.

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
