# Peer review of "Effects of Pair Versus Individual Housing on Performance, Health, and Behavior of Dairy Calves"

_animals, 2019, doi:10.3390/ani10010050_

Round 1

Reviewer 1 Report

Effects of pair versus individual housing on performance, health and behavior of dairy calves

The study seeks to determine the effects of pair versus individual housing of calves from birth through the weaning transition and to 10 weeks of age. The paper is timely as there is currently interest from dairy farmers, milk buyers and consumers in pair or group housing of milk fed calves.

The paper is generally well written, in a clear and concise style with limited grammatical and typographical errors. Clear hypotheses are stated. The methodology is generally well described although some additional detail with regards to the feeding of the calves (see below) would support the study. The data have been thoroughly analysed and presented. The results are thoroughly analysed and well presented.

Some specific questions and comments are listed below:

Introduction

Line 52 replace “…. repoted…” with "reported".

Line 56 – is “health check” referring to the method or frequency of assessing the health status of the calves? Please explain or use more familiar terminology.

Line 71 – This is the first time that the term “mixing” is used. Please explain or use more familiar terminology. The authors are referring to mixing of calves in group housing after weaning.

Line 75 – correct typos …….. the day to day changes in standing time……    

Materials and methods

Line 112 to 118 – how was the colostrum fed? What type of starter feed was offered (pellets or coarse mix)? Was any forage (straw or hay?) offered during the milk fed phase of the study?

Line 126 – TMR should be in full the first time used. What were the ingredients of the TMR? Give an overview of the ingredients.

Line 136 to 140 and Table 1 - Are the nutritive values presented in Table 1 the means of the results of analysis of samples collected each week? Or were the weekly samples bulked and then analysed.

Line 181 – 182 –references cited in the text (Chua et al., Abdelfattah et al., Overest et al.) don’t follow the same format as elsewhere in the paper.

Line 210 – typo “……duing..”

Line 228 – The section heading is “Growth” however the opening data are feed intakes. Suggest that the section heading is changed to better reflect the content.

Line 239 to 245 – The ADG data –the analyses that have been undertaken are unclear. Has the data set been analysed as a whole (week 1 to week 10) or have the data been analysed for the preweaning period (where there are 5 or 6 time points), during weaning (2 or 3 time points) and post weaning (1 or 2 time points) separately? This should be further explained in section 2.6.1.  What are the p-values for the repeated measures ANOVA? These should be stated on the graph or in the figure title. The difference between ADG in IH and PH calves in week 7 appears similar to that in week 4. However, this difference is not significant because of the effects of week 1, 2, 3 and 5. How has the p-value for week 7 and week 10 been determined – have post-hoc tests been used (which ones) or one-way ANOVA?

Line 251 – 253 (Table 3) – what time point do these data relate to? Are they means for the entire study period or means from the end of the study? Please clarify in the table title.

Line 260 – delete reference to effects of TP. They were measured 24h after birth. Some calves would not have moved to their housing until 48h.

Line 267 to 269 – The data in Figure 4 would be better presented as mean±sem in the Materials and Methods, Line 89, as passive transfer doesn’t relate to the hypothesis or treatments. This is useful, relevant descriptive information but not a key result.

Figure 5 Why is the profile for the lying times (Fig 5c) not the inverse of the standing times (Fig 5d), if the calves are not standing, they must be lying and vice versa. Standing times of IH calves increase from week 6 to week 8. However, those of PH calves fall before increasing, which is more in keeping with the lying data.

Line 337 – Is there a word missing in the sentence “………could get same performance (e.g. DMI……)”?

Line 357 typo “…….. no enough evidence……) “no” should be “not”

Line 357 – you may expect to observe more NCD in calves less than three weeks of age (see e.g. Bazeley 2003, In Practice), so the finding of more diarrhoea in PH calves may be more important that is being suggested in this part of the discussion. The incidence of diarrhea is lower in the IH calves during week 3, a benefit of individual housing? This should be stated in the conclusion.

Author Response

Dear reviewer 1,

   Thanks for your comments.

   We have thoroughly considered all comments and substantially revised our manuscript point by point. All issues raised in comments have been addressed to improve the manuscript.

    Kind regards,

    Liu Shuai

Reviewer 2 Report

animals-669607

This study is about the effects of pair versus individual housing on performance, health and behavior of dairy calves. Although this is a topic that has been studied quite intensively, the study provides some interesting results and confirms previous studies. The introduction can be improved on clarity. It lacks a good flow and logical order. Indicate also what the age of the calves is in the different studies in relation to weaning, post-weaning, mixing. That is important information to understand the statements.

Line 52: reported

Line 53: in a group

Line 57: provide a definition of weaning

Line 59: delete together

Line 60: why only for heifer welfare? Better to write calf welfare?

Line 61-62: unclear sentence, does not follow logically from previous sentences

Line 65: add space after pairs (check whole document for this)

Line 67-68: the objective is not clear in relation to previous part. You state that behavior has not been well characteriszed after weaning. However, in the objective you describe that you are going to study calves before weaning and after they were mixed (without specifying when this happens). In het hypothesis you mention weaning, post-weaning and mixing?

Line 68: paired versus individual

Line 71: What do you mean with from weaning to mixing?

Line 72: why a different diet? A different social environment is a consequence of mixing, delete after mixing

Line 74: Here you go back to weaning? Why is this relevant to mention here? It is about after weaning and about mixing.

Line 75: what is ‘the mixing period’?

Line 76: what do you mean with stage, period?

Line 77: what are mixing periods?

Line 78-79: explain why you hypothesize this

Line 88: 48 h instead of 2 d of age (to make it consistent with rest)

Line 91: delete days from the heading (d 1-42, etc)

Section 2.2.1: put all information with respect to one topic (e.g. feed and water provision) together

Line 99: after the birth

Line 104-108: confusing text because of double information. Please improve.

Line 111: why was chosen for sand as bedding? For young calves this is not insulating very well and therefore not very beneficial for their thermoregulation

Line 117: did the calves not receive any roughage, e.g. hay?

Line 128: delete ‘the’ before bedding

Line 137: write in full before you abbreviate terminology

Line 139: How did you do that in the paired-housed calves?

Table 1: explain abbreviations

Line 143: Body weights

Line 150: how did you record individual fecal samples in paired-houded calves and in the groups?

Line 161-167: when and how often were calves checked for respiratory diseases

Line 173: did you use 1 camera for all pens, or did you have multiple cameras?

Line 176: turn on instead of perform

Table 2: the definitions of behaviour are not mutually exclusive. A calf that stands and eats is active in a standing posture, but is it standing or feeding? Chewing and ruminating but also self-grooming can be done in a lying and standing posture, how did you observe this? What did you do with consuming milk? That is also a form of feeding. Replace non-nutrive sucking for non-nutrive manipulation of pen structures because that is what you describe. Replace fixtures and fitting for pen structures. What did you not observe important behaviours as walking, all forms of play behaviour, drinking water, etc.

Line 186: move ‘ all behavioral ....observer.’ to end of paragraph

Line189-191: please provide more information about the observations. How long, when etc.

Line 210: during

Line 212: behavioral

Line 213: why did you analyse weaning and post-weaning as one? Quite different stages.

Line 225: normally scientists use P<0.05

Line 231: you don’t have to repeat the weeks after weaning, post-weaning etc. if you have explained that in M&M

Line 232: paired-housed (please check whole document)

Line 233: replace ‘calves housed in pairs had ... intake’ for ‘than for individually housed calves’

Line 242: what do you mean with growth check?

Table 3: write in full before you abbreviate terminology, What age of the calves is this table about? Time effect is not very surprising with growing animals.

Line 262: what is TP?

Line 284-286: belongs to M&M

Line 290: that is obvious, when calves lie more the will stand less.

Line 295: more self-grooming than calves ...

Figure 6: huge figure, make smaller

Line 313: what is the difference between enhancement and social facilitation and learning?

Line 321: this is a very limited interpretation and from a production perspective. Social contact and housing has many more benefits from a welfare point of view.

Line 328: what about BW, at the time of mixing. It seems that the individual housed calves compensate for a previously experienced lack of social contact and related lower feed appetite or something like that.

Line 332: indicate the age of the calves of the studies you refer to. This is important to put results into perspective.

Line 346: please start with your own results and then put it into perspective with other studies.

Line 366-370: this part seems to better fit in the discussion on growth.

Line 371: why do you assume this competition for resources?

Line 375: is this not just a matter of increasing age? When calves grow older they have different behaviour and priorities, and a more complex social contact.

Line 387: Still very limited amount of observations to conclude from, be careful.

Line 388: self-grooming can also be an expression of stress, not knowing how to deal with the situation.

Author Response

Dear reviewer 2,

Thanks for your comments.

We have thoroughly considered all comments and substantially revised our manuscript point by point. All issues raised in comments have been addressed to improve the manuscript.

Kind regards,

Liu Shuai
